# Synthesis and Characterization of N-Methyl Fatty Hydroxamic Acids from Ketapang Seed Oil Catalyzed by Lipase

**DOI:** 10.3390/molecules24213895

**Published:** 2019-10-29

**Authors:** Dedy Suhendra, Erin Ryantin Gunawan, Hajidi Hajidi

**Affiliations:** Chemistry Department, Faculty of Mathematics and Natural Science, University of Mataram, Mataram 83125, Indonesia; dedysuhendra@unram.ac.id (D.S.); jirdycr7@gmail.com (H.H.)

**Keywords:** ketapang seed oil, N-methylhydroxylamine, N-MFHA, Lipozyme TL IM

## Abstract

N-methyl fatty hydroxamic acid (N-MFHA), which is a derivative of hydroxamic acid (HA), was synthesized from ketapang seed oil (*Terminalia catappa* L.). In general, HAs have wide applications due to their chelating properties and biological activities. N-MFHAs were synthesized using immobilized lipase (Lipozyme TL IM) in biphasic medium which was the ketapang seed oil dissolved in hexane and N-methylhydroxylamine dissolved in water. The products were characterized through color testing and FT-IR spectroscopy after purification. Various factors affecting the enzyme activity investigated in the study included the effect of incubation time, the amount of lipase used, and the temperature. On the basis of the results, the optimum conditions for the synthesis of N-MFHA obtained are 25 h of incubation time, a temperature of 40 °C, and a ratio of 1:100 for the amount of enzyme (g)/oil (g). At the optimum conditions of the reaction, 59.7% of the oils were converted to N-MFHA.

## 1. Introduction

According to [1], hydroxamic acid (HA), represented by the general formula, R-CO-NHOH, is a compound derived from oxoacid by replacing the –OH in it by –NHOH, and its hydrocarbyl derivatives. The HAs are much weaker acids as compared with the corresponding carboxylic acids despite having identical carbon chains, however, they possess stronger abilities for selectively chelating metal ions [2]. As a metal binding group, HA are also widely used as a metalloenzyme inhibitor, including tyrosinase inhibitors [3], histone deacetylase inhibitors [4,5,6,7], and HIV-integrase inhibitors [8]. Additionally, they have a wide spectrum of biological activities, such as anticancer [9,10], antituberculosis [11,12], anti-hepatitis C virus [13], and antibacterial [14] properties. Furthermore, the acid complexes with other metal ions are widely used in analytical chemistry, such as a reagent for gravimetric and spectrometric metals determination [15], as a chemical sensor in determination of metal ions [16], and as a reagent for removal of Acid Red 88 from textile wastewater [17].

On the basis of a literature review, it is evident that many studies have been carried out involving the production of HA from various feedstocks, which are now available as commercial products, however, the long chain HA have not been commercially available. They are widely used as surfactants in the pharmaceutical and soap industries [18]. Long chain HA are synthesized using fatty acids or vegetable oils as feedstock, in the presence of some enzymes. Fatty hydroxamic acids (FHA), one of the derivatives of long chain HA, have been synthesized with the use of both edible and non-edible vegetable oils. The edible oils that have been used as a raw material for the synthesis of FHA are soybean oil [19] and palm oil [20]. Meanwhile, the non-edible oil that has been used for the synthesis of FHA is ketapang kernel oil [21].

For this research, our focus is the synthesis of another FHA derivative, N-methyl fatty hydroxamic acids (N-MFHA), in the presence of lipase as catalyst. The ketapang seeds contain about 55% to 60% oil, which are basically medium to long chain of fatty acids. According to Suhendra et al. (2018) [21], the fatty acids composition of these seeds include palmitic (35% to 36%), palmitoleic (0.3% to 0.5%), stearic (4% to 5%), oleic (38% to 39%), linoleic (20% to 21%), arachidonic (0.5% to 0.6%), and others (0.1%). In addition, the ketapang trees are widely distributed in Indonesia and are among the plants which bear fruits throughout the year. Their oil also has some characteristic physical-chemical properties. Hence, it has the potential to be used as feedstock for production of the long chain HA under study, N-MFHA.

## 2. Results and Discussion

### 2.1. Identification of the Product

The identification of the *N*-MFHA, as shown in Scheme 1, was achieved by observing the color changes in the reaction of the products with copper (II), iron (III), and vanadium (V) metal ions. The formation of colored complexes between these metal ions and hydroxamic groups is a typical reaction (Suhendra et al., 2005). Green, dark red and purple are the complex colors of copper(II)-N-MFHA, iron(III)-N-MFHA and vanadium(V)-N-MFHA ions.

The FTIR spectrum of ketapang oil, as shown in Figure 1, has a typical absorption at 2950 to 2800 cm^−1^, which is C-H stretching from long chain alkyl. These C-H absorption bands are supported by peak at 1500 to 1450 cm^−1^ (C-H bonding) and the spectrum of product is a typical absorption band for hydroxamic acid groups, which peak at 3400 to 3200 cm^−1^. This is a typical absorption band for amide. The other absorption band at 1690 to 1630 cm^−1^ corresponds to the band for C=O of amide.

The nitrogen content of the product, which was determined using the Kjeldahl method, was 3.46%. This is an indication that there are 2.47 mmol N-MFHA in every 1 g of the product, however, the analysis of the products with HPLC shows that N-MFHA is a mixture of 36% palmityl-, 5% stearyl-, 30% oleyl-, and 29% linoleyl-*N*-methyl hydroxic acid.

### 2.2. Optimization of the Reaction Conditions

#### 2.2.1. Effect of Incubation Time

Enzymes are biocatalysts produced by living organisms for the purpose of accelerating chemical reactions, by reducing activation energy, but remain unchanged at the end of the reactions. They are the most abundant group of proteins in living cells and when a protein is experiencing denaturation, they lose catalytic activity over time. Some enzymes usually lose a large number of their catalytic activity during the incubation period. Hence, the effect of incubation time is a good indicator for enzyme performance and reaction progress. It pinpoints the shortest and most adequate time necessary to obtain good yields and minimize expenses [20].

As shown in Figure 2, the production of N-MFHA increases with increasing incubation time, up to 25 h. This is in line with the result of the study conducted by Salwanee et al. (2013) [22], who reported that the longer the incubation time, the more catalytic events had taken place at the active site, thereby resulting in higher amounts of N-MFHA. It is also evident from the figure that after 25 h, the amount of N-MFHA produced began to decrease. This is due to the fact that the reaction catalyzed by the enzyme is reversible. The incubation time obtained from this study is faster than the results of Suhendra et al. (2005) [20], during the synthesis of fatty hydroxamic acids from palm oil, which was at 30 h.

#### 2.2.2. Effect of Amount of Immobilized Lipase

In terms of production cost, the substrate concentration must be as high as possible to get higher yields, however, the amount of enzyme in the reaction should be as low as possible in order to obtain maximum product yield per unit of enzyme.

Considering Figure 3, the maximum production of N-MFHA occurs at a fixed ketapang seed oil concentration (0.12 g/mL), with the use of 0.02 g Lipozyme TL IM, after which the product yield is relatively constant. Increasing the amount of enzyme increases the rate of enzymatic reaction, until a limiting rate is reached, which is caused by more collisions between the enzyme and both the substrate and product molecules [21].

#### 2.2.3. Effect of Temperature

Generally, in all chemical reactions, an increase in temperature leads to an increase in the kinetic energy of the reacting molecules. Hence, there are more random collisions between the molecules per unit time, however, for enzymatic reactions, an increase in temperature also leads to an increase in the vibrational energy of the enzyme-substrate molecules.

As shown in Figure 4, the optimum temperature, at which the maximum reaction rate occurs, is 40 °C. The figure also shows an increase in temperature to the optimum, 40 °C, causing an increase in N-MFHA production, however, immediately after the optimum temperature, an increase in temperature causes a decrease in the production of N-MFHA. This is due to the fact that at the optimum temperature, the collision between the enzyme and substrate occurs very effectively, which causes the formation of an enzyme-substrate complex more easily. Additionally, according to [23], an increase in temperature increases the solubility of the substrate, thereby making available more substrates for the enzymatic reactions. Then, after the optimum temperature, the production of N-MFHA decreased quite sharply. This is due to the fact that at high temperatures, the active site of the enzyme changes shape, making it less complementary to the shape of the substrate, hence, its ability to catalyze is reduced. Finally, the enzyme is denatured and no longer functions [24].

### 2.3. Synthesis Using the Optimum Reaction Conditions

N-MFHA has been synthesized using the optimum conditions of the reaction, as presented in Table 1. The reaction conditions were 20 g of oil, 100 mmol of N-methylhydroxylamine, 150 mL of hexane, 0.2 g of lipozyme TL IM, 25 h of incubation time, and 40 °C.

## 3. Material and Methods

### 3.1. Material

All the chemicals and reagents were purchased from commercial suppliers and used as received. The researchers ensured that they were all of analytical grade and they included: *N*-methylhydroxylamine hydrochloride (Tokyo Chemical Industry, TCI, Tokyo, Japan), sodium hydroxide, methanol, and hexane from (Merck, Darmstadt, Germany), and the immobilized lipase (Lipozyme) from (Novozymes, Bagsværd, Denmark).

### 3.2. Extraction of Ketapang Seed Oil

The extraction of oil from the ketapang seeds was carried out using the modified method developed by Suhendra et al. (2018) [21]. First, the seeds were mashed and about 10 g were put into the Soxhlet extraction thimble and extracted with 250 mL of *n*-hexane for 6 h. Then, the extracts were purified by removing its solvent using a rotary evaporator at 40 °C and later the oil was passed into a chromatographic column containing silica gel, then eluted using *n*-hexane.

### 3.3. Synthesis of N-MFHA

The synthesis of N-MFHA from ketapang seed oil, in general, followed the modified procedures developed by Suhendra et al. (2018) [21]. This involved the addition of some ketapang oil with 15 mL of hexane in a 100 mL closed Erlenmeyer. Then, *N*-methylhydroxylamine hydrochloride, neutralized using 6 M NaOH, was added at a certain concentration. The mixture was incubated in a water shaker bath at a temperature of 30 °C to 40 °C with a shaking speed of 100 rpm. The N-MFHA formed in the hexane phase was separated from the water and Lipozyme TL IM phases. Then, the hexane phase was cooled to < 5 °C for five hours and the N-MFHA formed was filtered and washed using hexane to remove residual oil as well as other impurities. Finally, the N-MFHA was stored in a desiccator over phosphorus pentoxide.

### 3.4. Characterization

The qualitative analysis of the HA group was conducted using several modifications of the procedures developed by Suhendra et al. (2005) [20], mainly through color test and FTIR. Usually, a colored complex was formed whenever the methanolic solution of HA reacted with iron(III) or copper(II) ions. Then, the FTIR spectra of the products were recorded using a FTIR Spectrophotometer (Perkin Elmer FTIR–Frontier, Waltham, MA, USA). The quantitative analysis of the product was conducted by determining its nitrogen content using the Kjeldahl method. Additionally, the other quantitative analysis used high-performance liquid chromatography (HPLC, Waters, Milford, MA, USA). For HPLC, a Waters Model Breeze 1525 Preparative Gradient was also equipped with a Waters model 1525 binary pump, a Waters model 2489 UV/VIS detector, and a SunFire C18 5µm 4.6 × 150 mm reversed-phase column. The analysis was carried out using acetonitrile as eluent at a flow rate of 1.0 mL/minute and the absorbance monitored at 213 nm.

## 4. Conclusions

This study investigated the synthesis of *N*-methyl fatty hydroxamic acids from ketapang seed oil using the lipase-active biocatalyst Lipozyme TL IM. Various factors affecting the enzyme activity were investigated including the effect of incubation time, amount of Lipozyme TL IM used, and temperature. On the basis of the results, the optimum conditions for the synthesis of N-MFHA obtained are 24 to 26 h of incubation time, a temperature of 40 °C, and a ratio of 1:100 for the amount of Lipozyme TL IM (g)/oil (g).

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
