# Peer review of "Synthesis and Characterization of N-Methyl Fatty Hydroxamic Acids from Ketapang Seed Oil Catalyzed by Lipase"

_molecules, 2019, doi:10.3390/molecules24213895_

Round 1

Reviewer 1 Report

Some questions and corrections are presented in Pop-Ups of *pdf-file attached in zip.  The additional questions and comments are in *doc-file attached in zip.

Reviewer 2 Report

Suhendra and colleagues have investigated the synthesis of N-methyl fatty hydroxamic acids from ketapang seed oil using lipase as a catalyst. The optimum conditions appeared to be a temperature of 40 °C, an incubation time of 25 hours, and an enzyme : substrate ratio of  1 : 100 (w/w).

This is a somewhat “small” result. An important question has not been studied by the authors: How much product was obtained? Is this amount good enough for further exploitation, and perhaps commercialization? These data should be added.

Generally, the manuscript is sufficiently. Below I give some detailed suggestions for correcting the English and improving the clarity.

Line 12: is a derivative -> are derivatives

Line 13: could be -> were; Leave out “In general,”

Line 15: leave out “, such as a reagent”

Lines16- 17: HA are widely used as tyrosine inhibitor, anti-tuberculosis, antibacterial and HIV-integrase inhibitors. -> HA are widely used as anti-tuberculosis drugs, antibacterials, and tyrosinase and HIV-integrase inhibitors.

Line 19: at biphasic medium -> in a biphasic medium

Line 28: the hydroxamic acids -> hydroxamic acids

Line 30: please, add after “derivatives” “(see Scheme 1)”; weaker -> weaker acids

Line 38: determining -> determination

Line 43: are also be synthesized -> are synthesized

Line 46: that has been used as a raw material -> that have been used as raw materials

Line 49-50: synthesis of another derivatives of FHA, which is N-methyl fatty hydroxamic acids -> synthesis of other FHA derivatives, N-methyl fatty hydroxamic acids (N-MFHA),

Line 52: include; -> includes

Line 63: lead -> leads

Line 64: are with the copper -> when complexed with copper; ions. -> ions, respectively.

Lines 67-71 are duplicated in lines 72-76. Please, remove duplicate lines.

Line 68: supported -> accompanied

Line 69-70: Also, the spectrum of product is a typical absorption band for hydroxamic acid groups, which peaked at 3400-> The spectrum of the product contains a typical absorption band of the hydroxamic acid group at 3400

Line 79: which analyzed by -> as determined using

Line 81: palmity- -> palmityl

Line 86: reducing activation -> reducing the activation;  but remain -> but they remain

Line 88: there would loss in its catalytic activity with time -> there is loss of catalytic activity over time

Line 89: number -> amount

Line 90: in reaction -> of the reaction; Its pinpoints the shortest or -> It pinpoints the shortest or most

Line 96: which -> who

Lines 97-98: the more the proteolysis processes on the active side of the enzyme, thereby resulting in higher N-MFHA. -> the more catalytic events had taken place at the active site, thereby resulting in higher amounts of N-MFHA.

Line 99: N-MFHA -> the amount of N-MFHA

Line 100: no produce is formed yet -> no products have formed yet

Line 107: term -> terms; the concentration of the substrate -> the substrate concentration

Line 109: yield. -> yield per unit of enzyme, thereby reducing the cost.

Lines 133-137: this is speculation. At higher temperatures the enzyme is likely to unfold (or even denature), losing activity.

Round 2

Reviewer 1 Report

The main remark concerns the number of repetitions of experiments and statistical processing of the obtained results. No answer to the question is observed. What is the relative experimental error?

Typos are highlighted in blue in the text of attached file. Corrections are presented in Pop-ups

Author Response

Line 20......please forgive our limitations in English. You are right ...... additional experiments are needed to expand the reaction duration range ... we have done it using Response Surface Methodology ..... we will discuss it in our next publication.

Line 21 and 127-133 are implementations of reviewer 2's suggestion.

Thank you for all the suggestions  ....... very useful.

Best Regards

Dedy Suhendra et al

Reviewer 2 Report

I thank the authors for adding the summary data in paragraph 2.3 and Table 1.This answers my reservations.

I have one additional comment: please, add the main result to the abstract: At the optimum conditions of the reaction 59.7% of the lipids were converted intoN-MFHA. 

Author Response

As your suggestion, "At the optimum conditions of the reaction, 59.7% of the oils were converted into N-MFHA" has been added to the abstract